# Double Machine Learning Evaluation Under Distribution Shift and Selection Bias

## Abstract

Understanding how a model will perform when deployed in unseen environments is essential to preventing harm when algorithms inform decision-making. Two important drivers of model performance degradation are (i) *covariate shift* where the target covariate distribution differs from the source and (ii) *selective labels* where the observability of outcomes is influenced by the model itself. We study *pre-deployment* model evaluation under the joint presence of covariate shift and selective labeling. In particular, we present a double machine learning estimation procedure for the risk of an arbitrary black-box prediction model for a given loss function. We show identification of this estimand under standard assumptions, and derive a bias-corrected estimator based on the influence function of the target risk. We demonstrate our proposed estimator through controlled synthetic data and semi-synthetic eICU data experiments, which show that our estimator tracks the true target risk more accurately than combining standard plug-in approaches.

## 1 Introduction

Prediction algorithms that perform well within the training environment can degrade when deployed in new or changing environments. This degradation in performance is particularly consequential when the algorithms inform decisions that carry high-stakes and directly affect individual welfare or when the decisions induce changes in the environment. Moreover, the question of understanding performance degradation when deploying a model in environments that look different than the training data is inherently one of fairness: if left unaddressed, such models may disproportionately underperform for demographic groups that are underrepresented in the training data.

These concerns are well-supported empirically. As prediction algorithms are increasingly deployed to aid decision-making, evidence has mounted that performance can degrade significantly in new settings. For example, medical diagnosis algorithms have been shown to exhibit reduced performance for demographic groups that are underrepresented in the training data [42, 41, 14, 24, 30, 44]. Similarly, natural language processing tasks such as clinical text identification and hate speech detection often underperform on underrepresented subgroups and linguistic varieties [47, 31, 40, 29].

A prominent cause of degraded performance is *distribution shift* [34] where the training and deployment populations differ. One such class of distribution shift is known as *covariate shift* [43] where the distribution of input features changes while the causal relationship between features and the output remains constant. In particular, if the performance of the model varies across certain feature subgroups, covariate shift degrades model performance when the deployment population has a higher concentration of those features that are harder to predict. Even a model that performs well on average on the test set can have unpredictable real-world performance as a result of covariate shift [21].

A second pertinent source of performance degradation is when outcome labels are not observed uniformly at random across the population. In many settings, the observability of outcome labels

is determined by interventions that are themselves determined by the model's prediction. This phenomenon, referred to as the *selective labels problem* [23], impacts both learning and evaluation because of the selection bias that it imposes on the training data. It is difficult to estimate how the model would perform under counterfactual outcomes when the corresponding outcome labels are systematically missing.

These two challenges, *covariate shift* and *selective labels*, often coexist in practice when algorithms are used to aid in high-stakes decision-making. A salient motivating example is the use of *Clinical Decision-making Instruments (CDIs)* which are predictive models used in healthcare settings to assist with treatment assignment. CDIs use patient features including demographics, symptoms, and test results, to aid in diagnosis and treatment. CDIs trained on data from large, urban hospitals are deployed in rural communities where patient populations and medical practices look vastly different. Moreover, outcome labels are observable only for those patients for which the CDI indicated need for further testing or observation.

In this work, we address the task of pre-deployment model evaluation under covariate shift and selective labels. Our contributions are

1. We propose a target risk functional as an estimand to assess model performance in settings suffering from selective labels and covariate shift.

2. We demonstrate how to identify the target risk in terms of observable quantities in the data under a set of standard assumptions, and we characterize the influence function of our target estimand.

3. We construct a double machine learning estimator that requires access to only selectively labeled data from the source environment and unlabeled covariate data from the target environment. Our approach applies to arbitrary black-box prediction functions and general loss functions.

4. We empirically validate our method using synthetic experiments, and we illustrate our method in a real-world intensive care hospital setting.

## 1.1 Background and Related Work

**Dataset and Covariate Shift:** Here, we focus on *covariate shift* [43], where the marginal distribution of input features $P(X)$ changes between the training and deployment environments while the conditional distribution of the label given features $P(X|Y)$ remains unchanged[1]. Classic approaches to mitigating covariate shift rely on importance-weighted estimators [43, 46, 19], though such methods can suffer high variance. This challenge motivates the use of doubly robust methods for covariate shift correction [36, 15]. Beyond methods for correcting covariate shift, a growing body of works addresses the problem of evaluating models under covariate shift [7, 4, 2]. A related line of work examines whether a given shift is harmful in the first place, as not all shifts necessarily degrade performance [35, 32, 26].

**Selective Labels and Sample Selection Bias:** The *selective labels* problem arises when a model's predictions determine whether outcomes are observed. In such settings, outcome labels are available only for a biased sample of the overall population, which undermines learning and evaluation. In credit scoring mechanisms, an analogous challenge known as *rejection inference* is commonly addressed by training and evaluating models on only the labeled subset of samples [3, 1]; This approach has raised fairness and bias concerns [12, 13].

Another class of methods estimates the outcome for unlabeled samples [8, 5]. Alternatively, others leverage heterogeneity across decision-makers to correct the model and its evaluation [20, 6]. Other approaches include data augmentation procedures to acquire outcomes for subpopulations that are underrepresented in labeled samples [9] or directly incorporating consideration for downstream decision-making while training and evaluating models [11].

**Double Machine Learning** Double machine learning, also known as doubly robust estimation, is an estimation approach for settings with incomplete data that has become popular due to the desirable properties of the resulting estimators. In parametric settings, doubly robust estimators

---

[1]See, e.g., [34, 25, 28] for surveys on distribution shift more broadly.

remain consistent if either the propensity score or outcome model are correctly specified [38, 37, 22]. Such estimators enjoy fast rates of convergence in nonparametric settings [16] and have been used for estimation in settings closely related to selective labels and dataset shift, e.g., policy learning [10], covariate shift [7, 4, 2], and the challenge of data missing not at random [27, 45].

## 2 Problem Setting

We consider the problem of evaluating a fixed prediction model under the joint presence of covariate shift and selective labeling. Suppose that we observe $n$ independent and identically distributed (i.i.d) draws

$$Z := (X, R, RD, RY). \tag{1}$$

Each sample $Z$ comprises a covariate vector $X \in \mathcal{X} \subset \mathbb{R}^d$, a domain indicator $R \in \{0, 1\}$, a binary treatment $D \in \{0, 1\}$, and a scalar outcome $Y \in \mathbb{R}$. The label $R = 1$ designates units from the *source* population governed by law $P_S$ while the label $R = 0$ designates units from the *target* population governed by $P_T$; A binary treatment $D \in \{0, 1\}$ records an intervention of interest, and $Y$ is the associated outcome.

We adopt the potential outcomes framework [39] wherein each individual is associated with counterfactual outcomes $Y(1)$ and $Y(0)$ corresponding to the outcomes under treatment and no treatment, respectively. The observed outcome $Y$ is determined by the treatment assignment:

$$Y = D \cdot Y(1) + (1 - D) \cdot Y(0). \tag{2}$$

Due to selective labeling, $Y$ is only observed for units for which $R = 1$ and $D = 1$. In other words, we observe labeled outcomes only for treated individuals originating from the source distribution.

Let $P_S(X) := \mathbb{P}(X|R = 1)$ and $P_T(X) := \mathbb{P}(X|R = 0)$ denote the source and target covariate distributions, respectively, with corresponding probability density functions $p_S(x)$ and $p_T(x)$. We denote by $\mathbb{E}_S$ and $\mathbb{E}_T$ the expectation taken with respect to laws $P_S$ and $P_T$, respectively.

Our objective is to assess the accuracy of a fixed prediction model $f : \mathcal{X} \to \mathbb{R}$, which has been trained to estimate the treated potential outcome $Y(1)$. Specifically, we aim to evaluate the model under the target covariate distribution $P_T$. For a given loss function $\ell : \mathbb{R} \times \mathbb{R} \to \mathbb{R}_{\geq 0}$ (e.g., squared loss), the estimand of interest is the *target risk*:

$$\psi := \mathbb{E}_T \left[ \ell \left( f(X), Y(1) \right) \right]. \tag{3}$$

## 3 Identification and Estimation of the Target Risk

To describe the identification and estimation of $\psi$, it is convenient to introduce additional notation. We use $L := \ell(f(X), Y)$ as shorthand notation for the loss under $f$. Also define the following *nuisance functions*:

$$\pi(X) := \mathbb{P}(D = 1, R = 1|X), \quad \rho := \mathbb{P}(R = 0), \quad g(X) := \mathbb{P}(R = 0|X), \tag{4}$$

and

$$\mu(X) = \mathbb{E} \left[ L|X, R = 1, D = 1 \right], \tag{5}$$

the conditional mean loss among treated source units. Estimation of $\psi$ is complicated by the fact that $Y(1)$ is unobserved in the target domain ($R = 0$) and because, in the source domain, it is only observed for treated individuals ($D = 1, R = 1$). To ensure identifiability, we impose standard assumptions from causal inference and transfer learning:

**Assumption 1** (No unobserved confounding). $Y(d) \perp\!\!\!\perp D|X \quad \forall d \in \{0, 1\}$.

**Assumption 2** (Covariate Shift). $\mathbb{P}(Y(d)|X, R = 1) = \mathbb{P}(Y(d)|X, R = 0) \ \forall d \ \in \{0, 1\}$.

**Assumption 3** (Positivity). *There exists $\varepsilon > 0$ such that $\mathbb{P}(D = 1, R = 1|X) > \varepsilon$ almost surely.*

**Assumption 4** (Bounded density ratio). *There exists $C < \infty$ such that $\frac{dP_T}{dP_S}(x) \leq C \quad \forall x \in \mathcal{X}$.*

Assumptions 1-4 enable identification of the target risk (3) from observable data. Intuitively, these conditions require that (i) there are no unmeasured confounders, (ii) the relationship between covariates and outcomes is invariant across the source and target domains, (iii) every covariate profile admits a positive probability of treatment, and (iv) the source and target distributions' supports overlap sufficiently.

**Proposition 5** (Identification of the Target Risk). *Under Assumptions 1-4 and with nuisance functions $\pi, \rho, g$, and $\mu$ as defined in* (4) *and* (5)*, the target risk $\psi$ is identifiable from the observed data as*

$$\psi = \mathbb{E}_T\left[\mu(X)\right] = \mathbb{E}_S\left[\frac{p_T(X)}{p_S(X)} \cdot \frac{D \cdot R}{\pi(X)} L\right].$$

This result is the foundation of our proposed estimation procedure. The first equality provides a convenient expression of the estimand that enables our derivation of an influence function and the double machine learning estimator that it motivates. The second equality gives an alternative expression for our estimand that shows it can be identified from the source distribution by a reweighting procedure that resembles inverse propensity weighting (IPW) methods with an additional density ratio correction. A proof of this result is provided in Appendix A.1.

## 3.1 Estimation of the Target Risk

Next, we use the identification established in Proposition 5 to develop estimators for $\psi$ based on the functional's influence function. Let $\mathcal{P}$ denote the nonparametric model defined by Assumptions 1-4.

**Proposition 6** (Target Risk Influence Function). *For every $\mathbb{P} \in \mathcal{P}$, the map $\psi : \mathbb{P} \to \mathbb{R}$ admits the expansion*

$$\psi(\overline{\mathbb{P}}) - \psi(\mathbb{P}) = \int \varphi(z; \overline{\mathbb{P}}) \, d(\overline{\mathbb{P}} - \mathbb{P})(z) + R_2(\mathbb{P}, \overline{\mathbb{P}}), \tag{6}$$

*with influence function*

$$\varphi(Z; \mathbb{P}) = \frac{RD}{\pi(X)} \frac{g(X)}{\rho}(L - \mu(X)) + \frac{1-R}{\rho}(\mu(X) - \psi(\mathbb{P})). \tag{7}$$

*The remainder $R_2(\mathbb{P}, \overline{\mathbb{P}})$ comprises terms that are second order in the estimation errors of $(\mu, \pi, g)$ and first order in the estimation error of $\rho$.*

Proposition 6 establishes the influence function representation and remainder term expansion of the estimand $\psi$. The result follows from semiparametric efficiency theory, and is proved in detail in Appendix A.2.1. We outline the main steps of the proof here.

First, we identify a valid candidate influence function using established results on the influence functions of conditional expectations and densities (see, e.g, [18]). Next, we evaluate the efficiency of the candidate influence function by establishing an expansion of $\psi$ with respect to an arbitrarily perturbed distribution in $\mathcal{P}$.

## 3.2 Our double machine learning estimator of target risk

Motivated by the influence function derived in Proposition 6, we next construct a double machine learning estimator for the target risk $\psi$. Double machine learning, also known as one-step estimators or doubly-robust estimators, is a popular method for constructing estimators in settings with missing data such as causal inference [17]. To avoid overfitting due to nuisance parameter estimation, we employ standard sample-splitting techniques that retain the independence of nuisance parameter estimates by partitioning the data into independent folds. See Appendix B.1 for a detailed description of this procedure.

Formally, the estimator motivated by (7) is given by:

$$\widehat{\psi} = \frac{1}{n} \frac{1}{\widehat{\rho}} \sum_{i=1}^{n} \left[\frac{R_i D_i}{\widehat{\pi}_i} \widehat{g}(X_i)\left(L_i - \widehat{\mu}(X_i)\right) + (1 - R_i)\widehat{\mu}(X_i)\right]. \tag{8}$$

where $\widehat{\pi}, \widehat{g}$ and $\widehat{\mu}$ denote cross-fitted nuisance estimators, and $\widehat{\rho}$ is the empirical estimator of $\rho$.

The estimator (8) enjoys the *double robustness* property: In a parametric setting, it is consistent if either (i) the conditional mean $\mu(X)$ is correctly specified or (ii) the propensity score $\pi(X)$ and the density ratio $g(X)$ are correctly specified. If we are using non-parametric methods to estimate the nuisance functions, the estimator is $\sqrt{n}$-consistent and asymptotically normal under sample-splitting and $n^{1/4}$ convergence in the nuisance function estimation error.[2]

---

[2]This contrasts to standard methods like the plug-in or inverse probability weighting approach that would require $\sqrt{n}$ convergence in the nuisance function estimation error.

## 4 Experiments

### 4.1 Synthetic Experiments

**Synthetic Data Generation:** We evaluate and compare our proposed estimators via a synthetic experiment through a procedure that simulates the combined setting of covariate shift and selective labeling. We generate $n_S$ source samples $X_i^{(S)} \sim \mathcal{N}(\mu_S, \Sigma_S)$ and $n_T$ target samples $X_i^{(T)} \sim \mathcal{N}(\mu_T, \Sigma_T)$. By varying $\mu_S \neq \mu_T$ and/or $\Sigma_S \neq \Sigma_T$, we simulate covariate shift between source and target distributions.

**Treatment and Outcome Models:** For each source sample $X_i^{(S)}$, we compute treatment probabilities via a logistic regression model $\pi(X_i) = \sigma(\alpha^\top X_i)$, where $\sigma(\cdot)$ is the sigmoid function and $\alpha = c \cdot \mathbb{1}_d$ for a constant $c$ unless otherwise noted. We then sample treatment indicators $D_i \sim$ Bernoulli$(\pi(X_i))$, simulating a selection policy that depends on covariates. Then, we generate potential outcomes for the treated units in the source distribution $Y(1)_i \sim$ Bernoulli$(\text{sigmoid}(\beta^\top X_i))$ where $\beta \in \mathbb{R}^d$ is taken to be $C \cdot \mathbb{1}_d$ for an appropriate scaling factor $C$ unless otherwise noted. To introduce noise, we randomly flip the binary outcomes with probability $\gamma \in (0, 1)$, taking $\gamma = 0.1$ unless otherwise noted. Then we simulate selective labeling by setting $Y_i = \text{NA}$ for all units with $D_i = 0$, meaning outcomes are only observed for treated units.

**Model Training:** We randomly split the observed subset of the source data (i.e., units with $D_i = 1$) into 80% training and 20% test subsets. We train a logistic model on the training subset to predict the outcome $Y(1)$ from covariates $X$. This model $f$ is used to estimate $\mathbb{E}[Y(1)|X]$. We evaluate the mean squared error (MSE) of predictions on the held-out source test set.

**Nuisance Parameter Estimation:** To account for covariate shift, we fit a domain classifier (logistic regression) to distinguish between source and target samples, assigning the label $R = 1$ to the source and $R = 0$ to the target:

$$\widehat{w}(x) = \frac{1 - \widehat{\mathbb{P}}(R = 1|X = x)}{\max\{\widehat{\mathbb{P}}(R = 1), \varepsilon\}}$$

where $\varepsilon$ is a small positive constant to avoid division by zero. This yields an estimate of the density ratio $w(x) = \frac{p_T(x)}{p_S(x)}$. Next, we fit a logistic regression model to estimate the propensity score: $\widehat{\pi}(x) = \widehat{\mathbb{P}}(D = 1|X = x, R = 1)$ using logistic regression trained on the source samples which we evaluate on both the source and target data. Next, we compute the squared error losses $L_i = (f(X_i) - Y_i)^2$ for the subset of samples from the source distribution that are treated and have observable outcomes $Y_i$. Using these observed $(L_i, X_i)$ pairs, we train a random forest regressor $\widehat{\mu}(x)$ to estimate the expected loss $\mathbb{E}[L|X = x]$.

**Compute Naïve (Plug-in) Estimator:** The plug-in estimator computes the average predicted squared loss on the observed treated source data, reweighting by the estimated density ratio $\widehat{w}(x)$ and the inverse propensity weights $1/\widehat{\pi}(x)$ to account for both the covariate shift and selective labels. We compute: $\widehat{\psi}_{\text{plugin}} = \frac{1}{n_{RD}} \sum_{i=1}^{n_{RD}} \frac{R_i \cdot D_i \cdot \widehat{w}(X_i) \cdot L_i}{\widehat{\pi}(X_i)}$ where $n_{RD} = \sum_{i=1}^{n_P} R_i \cdot D_i$, the number of labeled samples in the source distribution.

**Compute DML Estimator:** We also compute the DML estimator for the target risk: $\widehat{\psi}_{\text{DML}} = \frac{1}{n} \sum_{i=1}^{n} \left( \frac{R_i \cdot D_i \cdot \widehat{w}(X_i)}{\widehat{\pi}(X_i)} (L_i - \widehat{\mu}(X_i)) + \frac{1 - R_i}{\widehat{P}(R=0)} \widehat{\mu}(X_i) \right)$ where $R_i$ is the domain indicator, $D_i$ is the treatment indicator, and $\widehat{P}(R = 0)$ denotes an empirical estimate of drawing from the target distribution.

**Estimate True Risk in Target with MC:** To estimate the ground truth target risk, we simulate an oracle dataset of $n_{\text{oracle}}$ samples from the target distribution. For each sample $X_i^{(T)} \sim \mathcal{N}(\mu_T, \Sigma_T)$, we generate a potential outcome using the same outcome model and again flip outcomes randomly with probability $\gamma$ to introduce noise. The ground truth risk estimate is then computed as the mean

squared error: $\psi_{\text{oracle}} = \frac{1}{n_{\text{oracle}}} \sum_{i=1}^{n_{\text{oracle}}} \left( f(X_i) - Y_i(1) \right)^2$ . This serves as a benchmark against which we evaluate our estimators $\widehat{\psi}_{\text{DML}}$ and $\widehat{\psi}_{\text{plugin}}$.

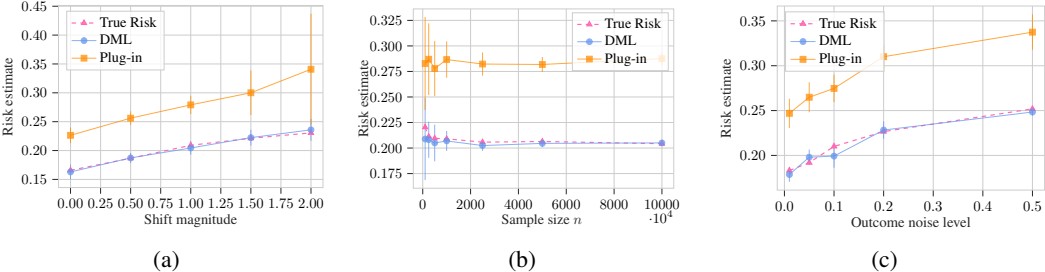

(a)            (b)            (c)

Figure 1: Results of synthetic experiment comparing DML and plug-in estimators for risk with the true risk under increasing covariate shift magnitude, increasing sample size, and increasing noise in the outcome model. (a) We increase the magnitude of $|\mu_S - \mu_T|$ along the direction of the true outcome model coefficients and report the average estimated risk over 30 trials for the plug-in and DML estimators, with error bars denoting standard deviations. (b) We increase the sample size $n = n_T = n_S$ and report the estimated risk averaged over 30 trials with error bars denoting standard deviations. (c) We increase the level of random outcome noise $\gamma$ (i.e., the probability of flipping the binary outcome of our outcome model) and evaluate the average estimated risk over 30 trials with error bars denoting standard deviations.

**Synthetic Experiment Results** In Figure 1a, we see that the DML estimator consistently tracks the true target risk more accurately across all covariate shift magnitudes where the shift is with respect to the mean of the Gaussian covariate distributions, while the plug-in estimator becomes increasingly biased as the shift grows. In Figure 1b, we see that the both the DML and plug-in estimators improve as sample size increases, while the DML estimator aligns closely with the true risk while the plug-in estimator appears biased. In Figure 1c, we see that both estimators capture the risk trend as outcome noise increases while the DML estimator once again tracks the target risk more accurately.

## 4.2 Semi-Synthetic Experiments

It is well-known that dataset shifts "in the wild" are often more complicated and difficult to address than shifts simulated in controlled, synthetic experiments [21]. This motivates experimentation that incorporates real covariates and identifies natural covariate shifts rather than simulating such shifts as our first set of experiments did. To accomplish this task, we access data from the eICU Collaborative Research Database [33] which includes intensive care unit (ICU) data from multiple treatment centers across the United States. We leverage the fact that the data include multiple treatment sites to simulate the setting where a model is trained on a population that differs in demographic makeup from the population on which it is to be deployed. By nature of the selective labels problem, we must still rely on the treatment and outcome models previously described in the fully synthetic experiment procedure since the data include only treated and labeled patients.

**eICU Data:** The eICU Collaborative Research Database [33] includes de-identified individual-level electronic health records from over 200,000 admissions to ICUs across multiple hospitals in the United States. Here, we focus on admission-level patient demographic and health data. We extract gender, ethnicity and age data, vitals including admission height, weight, and body mass index, clinical unit type (e.g., medical, surgical), and hospital ID. We one-hot-encode all categorical variables and impute missing values in continuous features with the median. All continuous features are standardized with Z-score normalization for computational tractability.

**Constructing Covariate Shift:** To simulate distribution shift that captures real-world complexity, we use patient data from a selected training hospital to construct the training environment and use all patient data from the remaining hospitals to construct the target environment. In particular, we select training hospitals that look systematically different from the general population in age and race/ethnicity. Figures Figure 2 and 3 compare the age and ethnicity covariate distributions of the identified source and target hospitals, respectively.

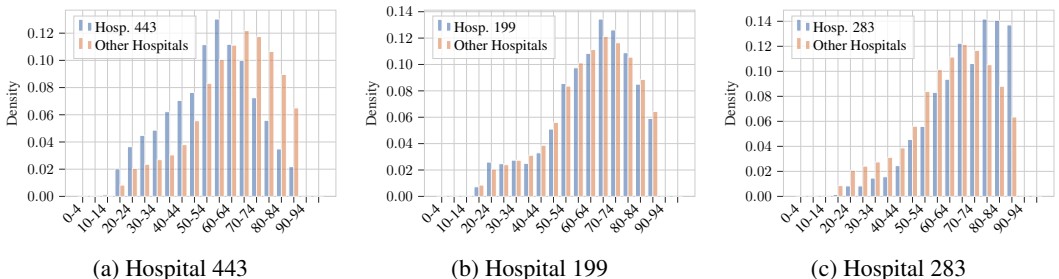

(a) Hospital 443       (b) Hospital 199       (c) Hospital 283

Figure 2: Comparison of age across hospitals in the eICU data. (a) Hospital 443 tends to have younger patients than other hospitals; (b) Hospital 199 has a typical age distribution; and (c) Hospital 283 tends to have older patients.

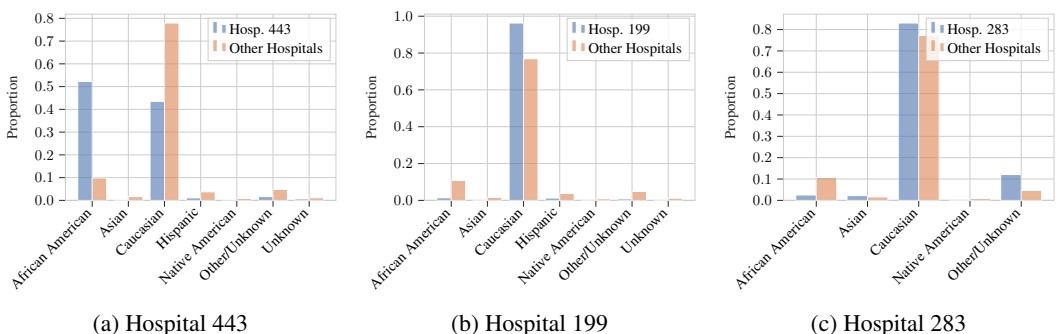

(a) Hospital 443       (b) Hospital 199       (c) Hospital 283

Figure 3: Comparison of ethnicity covariates across hospitals. (a) Hospital 443 tends to have more African American patients and fewer caucasian patients than other hospitals; (b) Hospital 199 tends to have more caucasian patients; (c) Hospital 283 tends to have a larger portion of patients with ethnicity unknown or labeled as "other".

**Semi-Synthetic Experiment Procedure:** Using actual patient covariates, we simulate models and outcomes by the same procedure as the purely synthetic experiments. We take $n_T$ to be the number of units in the training hospital. Then, we randomly select $n_T$ samples from the remaining hospitals to represent the unlabeled samples from the target setting. In other words, we take $n_S = n_T$. The rest of the samples are used to construct the oracle estimate of risk. Treatment is assigned using a draw from a Bernoulli distribution with probabilities determined by the patient features:

$$\pi(X) = \sigma(X^\top \alpha)$$

where $\alpha$ in this case takes a small constant $c$. The outcome is similarly generated via Bernoulli draws with probability determined by $X^\top \beta$ where $\beta \in \mathbb{R}^d$ is taken to be a randomly sampled and normalized vector of coefficients for each iteration. Once again, we simulate outcome noise by flipping a proportion $\gamma$ of the generated outcomes. The model fitting and estimator construction remains unchanged from the synthetic experiments. To estimate the true risk of deploying the model on the target population, we construct a Monte Carlo estimate of the risk using the remaining unused samples.

**Semi-Synthetic Experiment Results:** We use three different hospitals as the training center where each varies notably from the rest of the hospitals in its distribution of age, ethnicity, or both, as depicted in Figure 2 and Figure 3. We conduct experiments of the estimators under increasing noise in the outcome model as well as increasing propensity strength (increasing norm of $\alpha$) and increasing effect size (increasing norm of $\beta$). In Figure 4, we see that the DML estimator once again tracks the true risk more closely. Interestingly, here we observe behavior where the plug-in estimator both overestimates and underestimates the true risk. While *underestimation* of the true risk is particularly consequential in the medical contexts, *overestimation* is also relevant when data acquisition and model training are costly. In Figure 5, we see that increasing the propensity strength has little systematic effect on either estimator, though the DML estimator once again aligns with the

261 true risk more closely. Finally, in Figure 6, we see that increasing the effect size decreases the risk
262 estimate of both estimators as well as the true risk, where the DML estimator appears to increasingly
263 diverge from the true risk estimate under increasing effect size in Figure 6a.

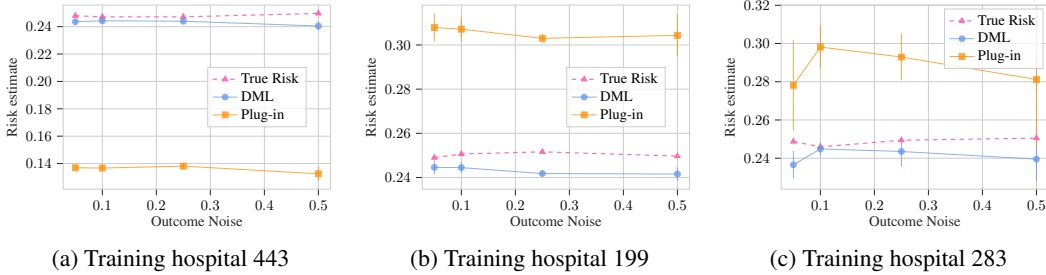

Figure 4: Comparison of DML and plug-in estimators with true risk across increasing noise levels in
the outcome model $\gamma \in (0.05, 0.5)$ across three different training hospital configurations. The error
bars represent standard deviation over 5 iterations. Our DML method more closely tracks the true
risk than the plug-in estimator.

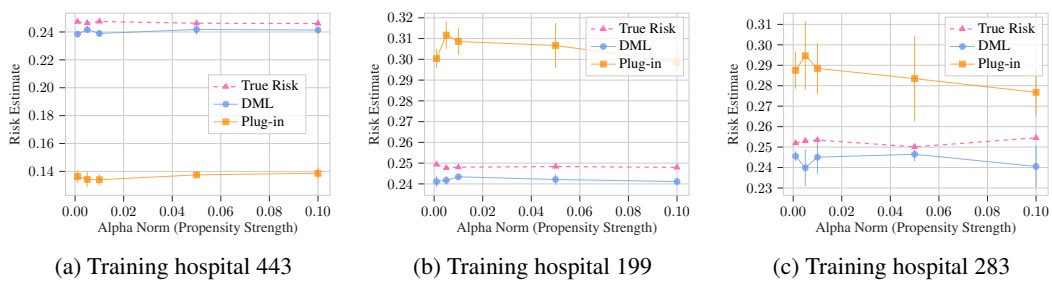

Figure 5: Comparison of DML and plug-in estimators with true risk across increasing norm in the
propensity parameter $\alpha$ and across three different training hospital configurations. The error bars
represent standard deviations over 5 iterations. Our DML method more closely tracks the true risk
than the plug-in estimator.

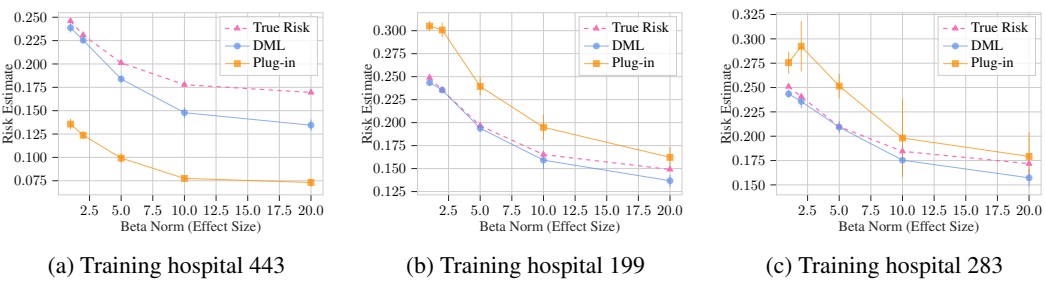

Figure 6: Comparison of DML and plug-in estimators with true risk across increasing norm in effect
size parameter $\beta$ and across three different training hospital configurations. The error bars represent
standard deviations over 5 iterations. Our DML method more closely tracks the true risk than the
plug-in estimator.

## 5    Conclusion

264

265 We studied the problem of pre-deployment model evaluation under the joint presence of covariate
266 shift and selective labels. We formalized the target risk as an estimand that captures a model's
267 expected performance in the deployment environment, and established conditions under which it
268 is identifiable from observable data. We derived an influence function representation of the target

risk and used it to construct a doubly robust, double machine learning estimator. Our estimator uses selectively labeled source data and unlabeled data from the target distribution.

Through synthetic and semi-synthetic experiments, we showed that our estimator more accurately tracks the true target risk in comparison with standard plug-in procedures. These results highlight the importance of developing tools that can account for multiple coexisting data challenges. In particular, the combination of covariate shift and selective labels, each of which has been studied extensively in isolation, poses distinct difficulties and is likely to arise in high-stakes domains such as healthcare.

Our work also points to several directions for future work. Relaxing the assumption of no unmeasured confounding and constructing similar estimators for other types of dataset shift would provide insight into other important domains where prediction algorithms inform decisions. In addition, many of the environments where our framework is relevant are also those in which it is natural to desire fairness-aware evaluation. Adapting our approach to explicitly consider fairness, e.g., by evaluating performance gaps across subgroups, would further strengthen the reliability of algorithmic decision-making.

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

# A  Proofs

## A.1  Proof of Proposition 5

Define $L^{(d)} \coloneqq \ell(f(X), Y(d))$ for $d \in \{0, 1\}$. We begin by showing the first equality. By law of total expectation,

$$\psi = \mathbb{E}_T \left[ L^{(1)} \right] = \mathbb{E}_T \left[ \mathbb{E} \left[ L^{(1)} | R = 0, X \right] \right]. \tag{9}$$

By Assumption 2,

$$\mathbb{E}_T\left[\mathbb{E}\left[L^{(1)}|R=0,X\right]\right] = \mathbb{E}_T\left[\mathbb{E}\left[L^{(1)}|R=1,X\right]\right]. \tag{10}$$

By Assumption 1,

$$\mathbb{E}_T\left[\mathbb{E}\left[L^{(1)}|R=1,X\right]\right] = \mathbb{E}_T\left[\mathbb{E}\left[L^{(1)}|R=1,D=1,X\right]\right] \tag{11}$$

By (2),

$$\mathbb{E}_T\left[\mathbb{E}\left[L^{(1)}|R=1,D=1,X\right]\right] = \mathbb{E}_T\left[\mathbb{E}\left[L|R=1,D=1,X\right]\right] \tag{12}$$

and the first equality follows by the definition of $\mu$ and by combining (9)-(12).

To show the second equality, we start with $\psi = \mathbb{E}_T\left[\mu(X)\right]$. We begin by expressing this quantity as an integral over the covariate space and applying Assumption 4:

$$\psi = \int_{x \in \mathcal{X}} \mu(x)p_T(x)dx = \int_{x \in \mathcal{X}} \mu(x)\frac{p_T(x)}{p_S(x)}p_S(x)dx = \mathbb{E}_S\left[\frac{p_T(X)}{p_S(X)}\mu(X)\right]. \tag{13}$$

By law of total expectation and by definition of $R$,

$$\mathbb{E}_S\left[\frac{p_T(X)}{p_S(X)}\mu(X)\right] = \mathbb{E}_S\left[\frac{p_T(X)}{p_S(X)} \cdot \mathbb{E}[L \mid X, R=1, D=1]\right] = \mathbb{E}_S\left[\frac{p_T(X)}{p_S(X)}\mathbb{E}_S[L \mid X, D=1]\right]. \tag{14}$$

Observe that, by another application of law of total expectation,

$$\mathbb{E}_S\left[L|X,D=1\right] = \frac{\mathbb{E}_S\left[DL|X\right]}{\mathbb{P}(D=1|X,R=1)}. \tag{15}$$

Combining (13)-(15) yields

$$\psi = \mathbb{E}_S\left[\frac{p_T(X)}{p_S(X)}\frac{\mathbb{E}_S\left[DL|X\right]}{\mathbb{P}(D=1|X,R=1)}\right].$$

An application of the Tower Property and the definition of conditional probability yields the claim. ∎

## A.2   Candidate Influence Function Derivation

The following lemma recalls well-known results characterizing the influence functions of conditional expectation and density functions. See, e.g., [17].

**Lemma 7** (Auxiliary Influence Functions). *For the conditional loss function $\mu(x)$, its influence function $\mathbb{IF}\left\{\mu(X)\right\}$ is given by:*

$$\mathbb{IF}\left\{\mu(x)\right\} = \frac{D \cdot R \cdot \mathbb{1}\{X=x\}}{\mathbb{P}(X=x,R=1,D=1)}\left(L - \mu(x)\right). \tag{16}$$

*Similarly, for the target covariate density $p_T(x)$, its influence function is given by:*

$$\mathbb{IF}\left\{p_T(x)\right\} = \frac{1-R}{\mathbb{P}(R=0)}\left(\mathbb{1}\{X=x\} - p_T(x)\right). \tag{17}$$

**Lemma 8** (Target Risk Influence Function). *Define*

$$\varphi(Z;\mathbb{P}) = \frac{RD}{\pi(X)}\frac{g(X)}{\mathbb{P}(R=0)}\left(L - \mu(X)\right) + \frac{1-R}{\mathbb{P}(R=0)}\left(\mu(X) - \psi(\mathbb{P})\right). \tag{18}$$

*Then $\mathbb{E}_{\mathbb{P}}\left[\varphi(Z;\mathbb{P})\right] = 0$ and, for every one-dimensional parametric sub-model $\mathbb{P}_\varepsilon = (1-\varepsilon) \cdot \mathbb{P} + \varepsilon\overline{\mathbb{P}}$ with score function $s_\varepsilon$,*

$$\frac{\partial}{\partial \varepsilon}\psi(\mathbb{P}_\varepsilon)\big|_{\varepsilon=0} = \mathbb{E}_{\mathbb{P}}\left[\varphi(Z;\mathbb{P})s_\varepsilon(Z)\right].$$

*That is, $\varphi(\cdot;\mathbb{P})$ is a influence function for $\psi$.*

### A.2.1 Proof of Proposition 6

Following the semiparametric calculus of [17], we treat $\mathcal{X}$ as a discrete set, apply Gateaux differentiation separately to each of $\mu(\cdot)$ and $p_T(\cdot)$, and invoke the product rule for influence functions:

$$\mathbb{IF}\{\psi\} = \sum_{x\in\mathcal{X}} \mathbb{IF}\{\mu(x)\}\, p_T(x) + \sum_{x\in\mathcal{X}} \mu(x)\mathbb{IF}\{p_T(x)\}.$$

Applying the building block influence functions (16) and (17) given in Lemma 7 together with Bayes' Rule, we obtain:

$$\mathbb{IF}\{\psi\} = \frac{R\cdot D}{\pi(X)}\frac{g(x)}{\mathbb{P}(R=0)}\left(L-\mu(X)\right) + \frac{(1-R)}{\mathbb{P}(R=0)}\left(\mu(X)-\psi\right).\blacksquare$$

### A.3 von Mises Expansion

**Lemma 9** (von Mises expansion). *For any two candidate laws $\mathbb{P}$ and $\overline{\mathbb{P}}\in\mathcal{P}$, the mapping $\psi:\mathcal{P}\to\mathbb{R}$ admits the expansion*

$$\psi(\overline{\mathbb{P}}) - \psi(\mathbb{P}) = \int \varphi(z;\overline{\mathbb{P}})\,d(\overline{\mathbb{P}}-\mathbb{P})(z) + R_2(\mathbb{P},\overline{\mathbb{P}}) \tag{19}$$

*where $\varphi$ is as defined in* (18) *and the remainder term $R_2(\mathbb{P},\overline{\mathbb{P}})$ is given by*

$$R_2(\mathbb{P},\overline{\mathbb{P}}) = \int \frac{\overline{g}}{\overline{\rho}}\overline{\mu}\overline{\mathbb{P}} - \int \frac{g}{\rho}\mu\mathbb{P} + \int \frac{\overline{g}}{\rho}\frac{\pi}{\overline{\pi}}(\mu-\overline{\mu})\mathbb{P} + \frac{\rho}{\overline{\rho}}\int\frac{g}{\rho}\overline{\mu}\mathbb{P} - \int\frac{\rho}{\overline{\rho}}\frac{\overline{g}}{\rho}\overline{\mu}\overline{\mathbb{P}}$$

*where we have supressed the arguments of functions in each term for brevity.*

*Proof of Lemma 9.* For any two candidate laws $\mathbb{P}$ and $\overline{\mathbb{P}}$ on $Z=(X,R,RD,RY)$, the von Mises expansion of the estimand $\psi$ around $\mathbb{P}$ is given by:

$$\psi(\overline{\mathbb{P}}) - \psi(\mathbb{P}) = \int \varphi(z;\overline{\mathbb{P}})\,d(\overline{\mathbb{P}}-\mathbb{P})(z) + R(\mathbb{P},\overline{\mathbb{P}}) \tag{20}$$

where $\varphi(z;\mathbb{P})$ is a candidate influence function of $\psi$ under $\mathbb{P}$ and $R(\mathbb{P},\overline{\mathbb{P}})$ is the remainder term which we will show is second-order. Since $\varphi(z;\overline{\mathbb{P}})$ is centered under $\overline{\mathbb{P}}$, (20) can be rearranged to express the remainder term as:

$$R(\mathbb{P},\overline{\mathbb{P}}) = \psi(\overline{\mathbb{P}}) - \psi(\mathbb{P}) + \int \varphi(z;\overline{\mathbb{P}})\,d\mathbb{P}(z). \tag{21}$$

To evaluate the remainder, we express the influence function in terms of the nuisance terms $\mu(X)$, $\pi(X)$, and $g(X)$ defined with respect to $\mathbb{P}$ together with their counterparts $\overline{\mu}(X)$, $\overline{\pi}(X)$ and $\overline{g}(X)$ defined with respect to $\overline{\mathbb{P}}$.

We make use of the following two identities which hold for any measurable functions $h(X,Y)$ and $h(X)$, respectively:

$$\mathbb{E}_{\mathbb{P}}[RD\cdot h(X,Y)] = \mathbb{E}_{\mathbb{P}}[\pi(X)\cdot h(X,Y)],$$
$$\mathbb{E}_{\mathbb{P}}\left[(1-R)\cdot h(X)\right] = \mathbb{P}(R=0)\cdot\mathbb{E}_{\mathbb{P}}\left[h(X)|R=0\right].$$

Applying these identities to our remainder term allows us to express the integral term as follows:

$$\int \varphi(z;\overline{\mathbb{P}})\,dP(z) = \mathbb{E}_{\mathbb{P}}\left[\frac{\overline{g}(X)}{\overline{\mathbb{P}}(R=0)}\frac{\pi(X)}{\overline{\pi}(X)}(\mu(X)-\overline{\mu}(X))\right] + \frac{\mathbb{P}(R=0)}{\overline{\mathbb{P}}(R=0)}\left(\int\overline{\mu}(X)d\mathbb{P}(X|R=0)-\psi(\overline{\mathbb{P}})\right). \tag{22}$$

By substituting the preceding integral term (22) into our expression for the remainder (21), we reach the expression:

$$R_2(\mathbb{P},\overline{\mathbb{P}}) = \int \frac{\overline{g}}{\overline{\rho}}\overline{\mu}\overline{\mathbb{P}} - \int \frac{g}{\rho}\mu\mathbb{P} + \int \frac{\overline{g}}{\rho}\frac{\pi}{\overline{\pi}}(\mu-\overline{\mu})\mathbb{P} + \frac{\rho}{\overline{\rho}}\int\frac{g}{\rho}\overline{\mu}\mathbb{P} - \int\frac{\rho}{\overline{\rho}}\frac{\overline{g}}{\rho}\overline{\mu}\overline{\mathbb{P}}$$

where we have suppressed the arguments from each nuisance function for compactness (i.e., we write $\mu$ for $\mu(X)$). A series of algebraic manipulations yield the equivalent expression:

$$R_2(\mathbb{P},\overline{\mathbb{P}}) = \int (\overline{\rho}-\rho)\frac{\overline{g}}{\overline{\rho}^2}\,\overline{\mu}\mathbb{P} + \int (\rho-\overline{\rho})\frac{g}{\rho\overline{\rho}}\mu\mathbb{P} + \int \frac{\overline{g}}{\overline{\rho}}\frac{(\pi-\overline{\pi})}{\overline{\pi}}(\mu-\overline{\mu})\mathbb{P} + \int \frac{(\overline{g}-g)}{\overline{\rho}}(\mu-\overline{\mu})\mathbb{P}.$$

$\square$

## B  Estimation Details

### B.1  Sample splitting nuisance function estimation

Given $n$ i.i.d. samples $\mathscr{Z}_n := \{Z_i = (X_i, R_i, D_i, Y_i)\}_{i=1}^n$ where each $Z_i$ is as in (1), we randomly partition the index set $\{1, \ldots, n\}$ into $K \geq 2$ disjoint folds $\mathcal{I}_1, \ldots, \mathcal{I}_K$ such that for each fold $k$, $|\mathcal{I}_k| \approx n/K$. For each index $i \in [n]$, let $k(i)$ denote the the fold containing the $i$-th observation. Then, for each fold $k$, construct an empirical estimate of each nuisance function using only samples outside of the $k$-th fold; Let $\widehat{\mu}^{(-k)}$, $\widehat{\pi}^{(-k)}$, and $\widehat{g}^{(-k)}$ denote such held-out estimates of the functions $\mu$, $\pi$, and $g$, respectively. Notice that, by construction, each of $\widehat{\mu}^{(-k)}$, $\widehat{\pi}^{(-k)}$, and $\widehat{g}^{(-k)}$ are independent of samples $Z_i \in \mathcal{I}_k$. Then, for each $i \in [n]$, set

$$\widehat{\mu}_i = \widehat{\mu}^{-k(i)}(X_i), \quad \widehat{\pi}_i = \widehat{\pi}^{-k(i)}(X_i), \quad \widehat{g}_i = \widehat{g}^{-k(i)}(X_i),$$

that is, evaluate the plug-in estimate on the held-out sample. To estimate $\rho$, we simply take the full sample mean:

$$\widehat{\rho} = \frac{1}{n} \sum_{i=1}^n (1 - R_i).$$

