# OpenReview forum: "Double Machine Learning Evaluation Under Distribution Shift and Selection Bias"
_NeurIPS.cc/2025/Workshop/Reliable_ML — NeurIPS 2025 - Reliable ML Workshop_

### Official Review · Reviewer_5PVV · 2025-09-18
**Nice paper using interesting tools**

**Rating:** 8
**Confidence:** 3

**Review:**

**Summary:**
- This paper uses tools from the doubly robust machine learning literature to develop an estimator for the risk of a block-box model that is robust to covariate shift and selective labeling. To build the estimator, they derive an influence function of the risk. They test their method against the naive plug-in IPW style estimator on several datasets.

**Strengths:**
- Overall, this submission represents an interesting contribution to the literature on doubly robust machine learning and learning in the presence of covariate shift / selective labels.
- It makes sense that it is a valuable and interesting contribution to study these two conditions in tandem rather than in isolation.
- The method is tested on both synthetic and partially synthetic datasets and the results show that the new method performs markedly better than existing plug-in methods
- The paper is well written on the whole.

**Weaknesses:**
- While it is acknowledged that real-world datasets are complicated to address in this setting, it would have been nice to see how the estimator performs in this environment.
- It's not clear whether this paper is testing against the "plugin" benchmark is the previous state of the art or whether there are stronger estimators to compare against.

**Suggestions:**
- The nuisance functions described on lines 115-117 should have more description about what each of these quantities represents.
- For proofs in the appendix, it is not fully clear which of these is already known and present in the literature versus a contribution of the project. What is Lemma 9 used for?
- It would be nice to provide a short description earlier in the paper about what estimators your estimator is competing against.
- It would be nice to have more explanation on the $n^{1/4}$ nuisance parameter convergence on line 167.

---

### Official Review · Reviewer_zCky · 2025-09-19
**Review of Submission 154**

**Rating:** 8
**Confidence:** 3

**Review:**

**Summary**

The paper studies pre-deployment risk evaluation under covariate shift and selective labels, aiming to estimate a model’s deployment-domain risk without target labels. It identifies the risk estimand under overlap and invariance assumptions, derives its influence function, and proposes a double machine learning (DML) estimator. The method is black-box and loss-agnostic, requiring only source labels and target covariates. Experiments on synthetic and eICU data show DML tracks true risk better than plug-in baselines.


**Strengths**

This paper presents a principled pipeline from identification to estimation that addresses a realistic, under-explored challenge in deployment. The DML estimator is broadly applicable (works with arbitrary predictors/losses) and empirically outperforms baselines across settings.

**Weaknesses**

Relies on strong assumptions (overlap, invariance, no unmeasured confounding) that may be hard to justify in practice. The paper offers limited guidance for checking these assumptions, making real-world use less clear.

**Suggestions**

Add a short “practical checklist” that explains how a practitioner might check whether the key assumptions (overlap, invariance, no unmeasured confounding) are at least plausible, and give simple tips for choosing or validating nuisance models. This would help bridge the gap between the strong theoretical assumptions and real-world deployment, making the method more usable beyond a theory audience.